# Acoustic Stress Induces Opposite Proliferative/Transformative Effects in Hippocampal Glia

**DOI:** 10.3390/ijms24065520

**Published:** 2023-03-14

**Authors:** Fernando Cruz-Mendoza, Sonia Luquin, Joaquín García-Estrada, David Fernández-Quezada, Fernando Jauregui-Huerta

**Affiliations:** Neuroscience Department, University Center of Health Sciences, University of Guadalajara, Sierra Nevada 950, Guadalajara 44340, Jalisco, Mexico

**Keywords:** hippocampus, Sholl analysis, microglia, astrocytes, stress

## Abstract

The hippocampus is a brain region crucially involved in regulating stress responses and highly sensitive to environmental changes, with elevated proliferative and adaptive activity of neurons and glial cells. Despite the prevalence of environmental noise as a stressor, its effects on hippocampal cytoarchitecture remain largely unknown. In this study, we aimed to investigate the impact of acoustic stress on hippocampal proliferation and glial cytoarchitecture in adult male rats, using environmental noise as a stress model. After 21 days of noise exposure, our results showed abnormal cellular proliferation in the hippocampus, with an inverse effect on the proliferation ratios of astrocytes and microglia. Both cell lineages also displayed atrophic morphologies with fewer processes and lower densities in the noise-stressed animals. Our findings suggest that, stress not only affects neurogenesis and neuronal death in the hippocampus, but also the proliferation ratio, cell density, and morphology of glial cells, potentially triggering an inflammatory-like response that compromises their homeostatic and repair functions.

## 1. Introduction

The hippocampus is a brain region that plays a crucial role in learning and memory, as well as the regulation of the Hypothalamus–Pituitary–Adrenal (HPA) axis. This region exhibits high levels of plasticity in adult mammals, marked by elevated amounts of cell proliferation, differentiation, and transformation. In addition to its role in memory, the hippocampus is connected to emotional and stress responses, receiving inputs from the amygdala and hypothalamus, key regulators of the HPA axis [1] (as reviewed in [2,3]). Moreover, the hippocampus is susceptible to the effects of chronic stress, expressing receptors for hormones such as corticotropin-releasing hormone (CRH) [4,5,6] (as reviewed in [7,8]) and glucocorticoids (GCCs) [9,10] (as reviewed in [2,11]).

Stress is a physiological response to environmental or physical challenges, that results in changes in various physiological parameters, including metabolism, immune response, and reproduction. To study the mechanisms and consequences of stress, researchers have used several experimental models. One such model is environmental noise (EN), which is considered an important pollutant and has been found to be a strong inducer of corticosterone (CORT) in rodents [12,13]. The impact of stress on the hippocampus has been well documented, with studies reporting a decrease in neurogenesis at the dentate gyrus (DG) and epigenetic modifications to the glucocorticoid receptor (GR) [14]. However, less is known about the effects of environmental stressors on glial cells, where adaptive responses may be expected [15].

Glial cells are the most abundant cell type in the brain. Astrocytes perform various functions such as axonal guidance [16,17], neurotransmitter recapture [18] (as reviewed in [19,20]), energetic metabolism (as reviewed in [20,21]), regulation of the blood–brain barrier [22], formation of a glial scar [23], and synapse integration [24]. In response to adverse stimuli, astrocytes undergo phenotypic transformation, differentiate into reactive astrocytes, release growth factors, and stimulate neurogenesis, angiogenesis, and promote the survival of neurons [25] (as reviewed in [26]). Microglia, the primary immune component of the brain, are closely associated with the inflammatory process. In response to damage, microglia exhibit three main activation states: (1) the resting state (M0), which serves as a microenvironmental sensor (as reviewed in [27,28]); (2) the classical proinflammatory activation state (M1), which produces proinflammatory cytokines and acts as an enlarged, unbranched macrophage, that eliminates damaged tissue, debris, or infection [29,30]; and (3) the alternative activation phenotype (M2), which is the opposite state of M1, produces anti-inflammatory cytokines, and restores the homeostatic state after a noxious event. M2 has a shape that is closer to M0, but more branched (as reviewed in [31,32]).

Whether chronic stress may have a harmful effect on the hippocampus, we aim to explore cell proliferation and glial transformation, to better understand how environmental stressors may impact these regenerative and adaptive capabilities. To answer this question, we have measured the number of cells proliferating after exposure to stimuli, evaluated the percentage of newly generated cells that correspond to microglia or astrocyte phenotypes, and performed a morphometric comparison of these phenotypes, using Sholl analysis. We have chosen environmental noise as an experimental model, due to its inescapability and prevalence in human environments.

## 2. Results

### 2.1. Effect of Environmental Noise on Cell Proliferation in Hippocampal Subregions

To examine changes in hippocampal cytogenesis, we used the proliferation marker 5′bromodeoxiuridine (BrdU), and administered it to the animals. The injections were given once the experimental subjects had completed their first week under noisy conditions. The animals were then sacrificed two weeks later, in order to conduct a thorough evaluation. We found that the number of new-born cells (BrdU-positive) that survive two weeks after injections, were significantly diminished in rats exposed to environmental noise. This was determined through a one-way ANOVA, which showed a significant difference in the number of hippocampal BrdU-positive cells (*F*_(7, 392)_ = 22.55, *p* = 0.0001). Specifically, a decrease in the proliferation rate was noted in CA1 (*p* = 0.0001), CA2 (*p* = 0.0001), CA3 (*p* = 0.0001), and DG (*p* = 0.0001) (Figure 1). 

### 2.2. Impact of Environmental Noise on the Newly Differentiated Astrocyte and Microglial Cells in the Hippocampus 

Newly differentiated glial cells were identified by double-labelling the new born-cells (BrdU+) that differentiated to glia, with either the phenotypic marker GFAP, for astrocytes, or IBA-1, for microglia. We found that environmental noise reduced the number of proliferating astrocytes (BrdU+/GFAP+) in the hippocampus (*F*(_7, 328_) = 13.25, *p* = 0.0001) (Figure 2A). The number of BrdU+/GFAP+ cells decreased in every subregion of the hippocampus (CA1, *p* = 0.0095; CA2, *p* = 0.0001; CA3, *p* = 0.0095; and DG, *p* = 0.0095). 

Additionally, there was also a significant difference in the number of proliferating microglia (BrdU+/IBA-1+ cells) (*F*_(7, 570)_ = 19.38, *p* = 0.0001) (Figure 2B). The noise-exposed group, increased the number of BrdU+/IBA-1+ cells in CA1 (*p* = 0.001), CA2 (*p* = 0.001), CA3 (*p* = 0.001), and DG (*p* = 0.0440), compared to the control group.

### 2.3. Effect of Environmental Noise on the Astrocyte Morphometry 

Sholl analysis was used to evaluate the morphometric complexity of astrocytes, by assessing the number and length of processes. In CA1, the number of intersections decreased significantly as a consequence of noise (*t* = 4.881; *p* = 0.0001). These reductions were significant for the number (*t* = 10.50; *p* = 0.0001), but not for the length, of processes (*t* = 1.957; *p* = 0.0515). In CA2, the number of intersections increased significantly in the exposed group, in comparison to the control group (*t* = 6.036; *p* = 0.0001). The length of processes also increased in the noise group (*t* = 4.432; *p* = 0.0001). There were no significant differences in the number of processes (*t* = 1.571; *p* = 0.1182). In CA3, the number of intersections decreased significantly in the exposed group, in comparison to the control group (*t* = 2.394; *p* = 0.0204). The number of processes also decreased (*t* = 4.193 *p* = 0.0001). However, the length of processes increased significantly (*t* = 4.273; *p* = 0.0001). In DG, the number of intersections also decreased significantly, as an effect of environmental noise (*t* = 10.48; *p* = 0.0001). The number of processes decreased (*t* = 5.490 *p* = 0.0001), but the length of processes did not exhibit any significant difference (*t* = 1.515; *p* = 0.1311). These results are summarized in Figure 3.

### 2.4. Effect of Environmental Noise on the Morphometry of Microglia in Hippocampal Subregions

In CA1, the number of intersections decreased significantly in the noise group compared to the control group (*t* = 9.801; *p* = 0.0001). These reductions were significant for the number of processes (*t* = 1.885; *p* = 0.0304) but not for the length of processes (*t* = 1.025; *p* = 0.3063). In CA2, the number of intersections decreased significantly as a consequence of stress (*t =* 14.88; *p* = 0.0001). Those reductions were significant for the number of processes (*t =* 4.512; *p* = 0.0001) although not for the length of processes (*t =* 0.2074; *p* = 0.8359). In CA3, the number of intersections decreased significantly in the environmental noise group in comparison to the control group (*t* = 14.62; *p* = 0.0001). The number of processes also decreased (*t* = 4.273 *p* = 0.0001), however, the length of processes increased significantly (*t* = 0.2887; *p* = 0.7731). In DG, the number of intersections decreased significantly in the exposed group compared to the control (*t =* 11.19; *p* = 0.0001). The number of processes decreased (*t =* 3.577; *p* = 0.0004) but their lengths did not exhibit any significant difference (*t* = 0.1.515; *p* = 0.1311). These results are summarized in Figure 4.

### 2.5. Density of Astrocytes and Microglia in Hippocampus after Environmental Noise

Except for the CA1 region, the exposed rats had a lower density of astrocytes in all other subregions of the hippocampus (CA2, CA3, and DG), as shown by the statistical analysis (*F*_(7, 228)_ = 24.93, *p* = 0.0001) (Figure 3A). Specifically, the density of astrocytes was lower in CA2 (*p* = 0.0131), CA3 (*p* = 0.0001), and DG (*p* = 0.0001), compared to the control group Figure 5A. In terms of microglia, there were no significant differences with respect to the control group (*F*_(7, 232)_ = 1.014, *p* = 0.4223) Figure 5B.

## 3. Discussion

Our study aimed to investigate whether acoustic stress can influence hippocampal proliferation and glial cytoarchitecture in adult male rats. We evidenced that new-born cell survival was abnormal after 21 days of noise stimulation. Besides the expected reduction in cellular proliferation, we observed an inverse/opposite effect on the proliferation ratios of astrocytes and microglia (GFAP+/BrdU+ and Iba1+/BrdU+ cells). Additionally, both cell lineages showed an atrophic morphology, with fewer numbers and lengths of processes. Lastly, we observed less astrocytic density in the stressed animals, while no changes were found in the microglial population.

Stress and cytogenesis, are two adaptive responses that help the body to cope with a variety of stimuli or damage. While in the former, the stimulus can induce damage by forcing the stress response to actively compensate, until it becomes unsustainable or expends so many resources that it compromises other functions (reviewed in [33]), the latter is presented as a way to repair damage, by modifying proliferation, differentiation, and cell death/survival rates. In some cases, however, these responses can become imbalanced and lead to long-term damage (reviewed in [34]). So, looking for a stimulus that is chronically annoying for humans, we chose a stress model that is not easily avoided, ubiquitous, and hard to habituate, even without strident conditions. In order to better extrapolate its impact on humans, our model included urban noises, that were adapted to the rat audiogram [35]. This kind of model has the advantage of being representative of the environmental conditions that humans are often exposed to, and, due to the difficulty of avoiding or adapting even at low levels, it allows a comparison of the stress responses of humans to those of other species.

First, our experiment evaluated changes in hippocampus proliferation, as a consequence of the chronic exposure. Our findings suggest a general decrease in the number of cells that proliferate and that survived at the end of the experiment. As seen in our results, the physiological cell repopulation in this region was interrupted by chronic exposure to noise. This change could be part of the commonly observed phenomenon in chronic stress models, that report a decrease in hippocampal volume [36,37], which, by some authors, has been attributed to a lower neurogenesis and the consequent reduced ability to replace lost cells [38,39,40]. To better understand the cell types involved in this decrease, we then analyzed the most abundant and closely related cell phenotypes to the repair or prevention of tissue damage, namely astrocytes and microglia. Interestingly, we found a differential modification that affected these two strains oppositely; while astrocyte repopulation saw a significant decrease, microglial proliferation was notably increased. Both phenomena are interesting to analyze, considering that they are a result of exposure to a stimulus (environmental noise) that is commonly, and often unwittingly, present in our lives. We already introduced the importance of both cell types from the physiological and pathological points of view. Astrocytes are responsible for a large number of functions, that are vital for the proper functioning of the hippocampus. Consequently, the deficit resulting from their low repopulation could translate into changes in cognitive performance [39], reduced homeostatic capacity of the region, or be part of the aforementioned phenomenon of hippocampal atrophy [41,42]. It is important to note, that some of these cells, specifically those located in the subgranular layer of the dentate gyrus, are b-precursor cells, and that the specific decrease in these cells affects the replenishment of precursors in the hippocampus, the major neurogenic niche of the adult brain [43]. Support for our finding has been reported by studies noting that astrocytes can be reduced under conditions of chronic [13,42], but not acute stress [44]. Thus, the reduction of astrocytic proliferation in the hippocampus could be interpreted as a phenomenon of chronicity. On the other hand, increased proliferation of microglia is considered an indicator of neuroinflammation [45], that has been observed to impact physiological roles, such as neurogenesis and cell survival [46,47] (reviewed in [48]). Our results, indicating an increase in microglial proliferation, are in line with previous studies, that also report an increase under aversive conditions [46,49]. However, one could expect the opposite response, if considering that steroids such as dexamethasone (a glucocorticoid analogue, GR agonist) have anti-inflammatory effects (reviewed in [50]). Contrary to this, our results suggest that the stressful stimulus used in this experiment may lead to an inflammatory-like response, if maintained for prolonged periods of time (reviewed in [51]).

The Sholl morphological transformation analysis, showed that astrocytes had fewer branches and fewer intersections, while microglia showed a type M1 phenotypic morphology, with fewer branches and a reduced number of intersections [30] (reviewed in [29]). Changes in astrocyte morphology have typically been associated with reactive gliosis, characterized by hypertrophic morphology, which usually appears after injury or pathological state [52,53] (reviewed in [54]). In our experiment, however, the results show that morphological changes go in the opposite direction, exhibiting atrophy. Although astrocyte transformation is closely related to post-lesion damage, this change could also reflect a mechanism that allows the hippocampus to prioritize certain functions, and not necessarily a failure by itself (reviewed in [55,56]). Given that astrocyte functions are closely related to their morphology and cell contacts, pathological interpretations might be considered based on the lost/acquired functions [55,57], which have been described as stimulus-dependent [52]. Note that the GFAP immunohistochemical analysis we used for morphometric analysis, could only signify a rearrangement of its cytoskeleton or its cellular processes [55]. In regards to microglia, our morphometric analysis aligns with the inflammatory profile described in previous studies [58,59] (reviewed in [48,60]). This transformational profile is associated with increased phagocytic index [61,62], greater production of pro-inflammatory cytokines [63], alteration of synaptic pruning patterns [64], and neuronal apoptosis [65]. While our data seems to match these phenomena, we should mention that some authors have proposed that, chronically activated microglial cells shift to an anti-inflammatory state that improves survival rates of the remaining cells after injury [66] (reviewed in [67]). Thus, we can expect that microglia may remain in a chronic activation state, that detracts from their ability to manage threatening stimuli. If observed in combination, the reduction shown by astrocytes and their processes could also be impacting the microglial activation phenotype observed here [68].

Finally, we investigated the effect of proliferation ratios and the microglial inflammatory profile, on the cellular densities of astrocytes and microglia. In line with our results on proliferation and transformation, we found that astrocyte density was decreased in most of the evaluated hippocampus subregions. This may suggest a decrease in cell-to-cell interactions, a reduction in modulatory and compensatory astrocytic capabilities, an accumulation of synaptic neurotransmitters in intercellular spaces [69,70] (as reviewed in [71]), an imbalance in Ca++-dependent glutamate release [72], alterations in the permeability of the blood–brain barrier [42,46,73,74] (as reviewed in [71]), and a reduction in memory process acquisition [75]. Previous research has shown, that memory and other hippocampus-mediated cognitive processes are reduced under stress conditions, so our results suggest that astrocyte dysfunction may contribute to these phenomena [76,77,78]. On the other hand, microglial cell density was unchanged, even with a higher proliferation ratio. This could mean that, unlike astrocytes, microglial cells die and proliferate in a more balanced manner. There is evidence to suggest, that the main stress mediator, CRH receptor type 1 (CRH-R1), could be involved in this effect. Previous experiments have reported that exposure to CRH increases CRH-R1 expression and induces proliferation in microglia [79]. Additionally, evidence exists, showing that this receptor–ligand activity induces apoptosis in microglia, particularly in activated cells [80]. Thus, the stress mediator CRH could act as a control mechanism for the microglia population, to reduce the pro-inflammatory state and re-establish homeostasis.

Our study provides important insights into the effects of noise exposure on glial morphology, but it is clear that noise exposure does not occur in isolation. There are many other conditions that can potentiate or interact with noise to produce a range of physiological or pathological outcomes. For example, sleep deprivation, which could be induced by persistent noise, has been shown to promote brain inflammation, glial cells activation, and changes in cellular composition of the hippocampus [81,82,83]. It is possible that conditions like sleep deprivation, housing conditions, or social stress, interact with noise, to produce synergistic effects that are greater than the sum of the individual effects. Future studies should explore these interactions in more detail, to gain a better understanding of the complex relationship between noise and the hippocampus.

To summarize, our results indicate that stress could affect, not only neurogenesis and neuronal death in the hippocampus, but also the proliferation ratio, cell density, and morphology of astrocytes. Stress may also trigger an inflammatory-like response in hippocampal microglia, compromising their homeostatic and repair functions. To our knowledge, this study is the first to evaluate the proliferative and morphometric changes in the two most abundant glial cells in the brain. Further research is needed to evaluate the tissue stiffness changes, activation state markers, and/or cytokine release in these cells, in order to gain a deeper understanding of the glial transformation and dynamic changes that can persist as long as the stressor persists.

## 4. Materials and Methods

### 4.1. Animals 

In this study, 90-day-old Wistar male rats were used. A total of 16 male rats, weighing 305.2 ± 30.6 g, were randomly divided into two groups: a control group (*n* = 8) as a non-exposed group, and a chronic environmental noise group (*n* = 8). In both groups, the animals were distributed four per cage, to avoid overcrowding. All groups were maintained in a 12:12 light–dark cycle, with lights on at 6:00 am. The temperature in the experimental room was maintained at 25 ± 2 °C, and humidity at 70%. The animals had free access to tap water and balanced food. The cages were changed in the testing room. All animal experiments followed the National Institute of Health guide for the care and use of laboratory animals, and the study was approved by the Health Sciences Ethics, Biosafety, and Scientific Board, at the University of Guadalajara, México. Figure 6 illustrates all experimental procedures.

### 4.2. Noise Exposure

Chronic environmental noise in this study, consisted of representative sounds of urban environments (i.e., turbines, hooters, horns, and others), adapted to the rat audiogram, as described by Rabat [35]. The sounds played were tailored to the rats’ lower capacity to detect low frequencies (under 500 Hz), and to perceive high frequencies (over 8000 Hz). Metal grid cages were used, to avoid sound reverberation, and the animals were housed in groups of four, in a sound-proofed room. Professional tweeters (Yamaha, Inc. Osaka, Japan) were placed 1 m above the cages and connected to a Mackie amplifier (Mackie M1400; freq. 20 Hz to 70 kHz; 300 W at 8 Ω). The speakers’ characteristics and the tweeter, allowed the sounds to be reproduced at frequencies between 20 and 50,000 Hz. Recordings of noisy environments were played randomly, with noise exposures (18–39 s of turbine, hooter, or horn sounds) and intervals of silence ranging from 20 to 165 s. Sound intensities were presented in a range of 70 to 105 dB, to avoid cochlear damage. This protocol was followed for 21 days (chronic group). A sound level meter was placed at every corner of each cage, to ensure that all the animals received the same exposure dose. To avoid housing confounds, all the rats were transferred to the testing room 48 h before the start of the stimuli.

### 4.3. BrdU Injection

The thymidine analog BrdU (Sigma-Aldrich, St. Louis, MO, USA), was dissolved in 0.9% NaCl and prepared at a concentration of 100 mg/mL. BrdU injections were administered during the light phase of the cycle, on days 7, 8, and 9 of the experiment. Each animal received three separate injections of their assigned dose, with injections given at approximately 9:00 a.m., 11:00 a.m., and 1:00 p.m. Solutions were slowly warmed and stirred on an ultrasonic homogenizer, until BrdU was dissolved, and then injected intraperitoneally (i.p.), as shown in Figure 6.

### 4.4. Perfusion 

The rats received an intraperitoneal injection of sodium pentobarbital (60 mg/kg), and were perfused through the left cardiac ventricle with 150 mL of saline solution, followed by 200 mL of 3.8% paraformaldehyde in 0.1 M phosphate buffer saline (PBS), pH 7.4. After perfusion, brains were removed and sectioned into coronal slices (35 μm), using a vibratome Leica VT1000E. Coronal sections were obtained under observation, stereoscopically. Blocks of tissue contained the regions of Bregma 3.72 mm to Bregma −4.68 mm. The posterior analysis was focused on the hippocampus (GD, CA1, CA2, and CA3).

### 4.5. Immunohistochemistry 

We performed a double immunofluorescence staining method, to colocalize cells that underwent mitosis (BrdU+) with the filamentary protein expressed throughout the cytoskeleton of astrocytes, glial fibrillar acidic protein (GFAP), and ionized calcium-binding adapter molecule 1 (Iba-1), a microglia response factor, in the hippocampal dentate gyrus (DG) and Cornu ammonis (CA1, CA2, CA3) regions.

The tissue sections were incubated in 2N HCl for 20 min, at 37 °C, and rinsed with 0.1 M borate buffer (pH 8.5). Then, the slices were washed with phosphate buffer (PBS) and incubated with a blocking solution, 5% normal goat serum (NGS) plus triton 0.1%, in 0.1 PBS, for 30 min, at room temperature. Slices were incubated in primary antibodies in the blocking solution for 18 h, at 4 °C, that contained a monoclonal α-BrdU rat (1:750 Ab-cam, CAT: ab6326), α-GFAP rabbit (1:750 Dako, CAT: Z0334), and α-Iba-1 rabbit (1:250 FUJIFILM Wako Pure Chemical Corporation, CAT: 016-26461). After rinsing in PBS, the sections were incubated for 2 h with the secondary antibodies (Alexa Fluor 488 goat anti-rat IgG and Alexa Fluor 594 goat anti-rabbit from Molecular Probes at 1:1000 dilutions). The sections were mounted on slides and cover-slipped using fluorescent mounting media (Vectashield Vector Labs, Burlingame, CA, USA). The sections from experimental and control rats were processed simultaneously, to ensure access to the same sets of solutions.

### 4.6. Quantification Density of GFAP^+^ and Iba-1^+^ Glial Cells 

The stained tissues from all the groups were photographed with a Leica DFC 7000T camera, attached to a DMi8 microscope. We counted the GFAP+ and Iba-1+ glial cells on an average of eight to ten coronal slices. Both hemispheres were quantified for every region expressing the marker. The 20× objective was used to obtain photographs per region. No adjustments (contrast, intensity, or gamma correction) were applied to the images. Every photograph represented a microscopic field, in which the dimensions of each region were completely covered. Image analysis was performed using the ImageJ software and the plug-in: Cell Counter Notice (https://imagej.nih.gov/ij/, ver.1.44, National Institutes of Health, Bethesda, MD, USA, accessed on 15 June 2022). The results were averaged and compared between the groups.

### 4.7. Analysis of Proliferation and Differentiation

The proliferation of newborn cells was counted manually, using the ImageJ software and the plug-in: Cell Counter Notice (https://imagej.nih.gov/ij/, version 1.44, National Institutes of Health, Bethesda, MD, USA). The differentiation to astrocyte or microglia was analyzed through the percentage of colocalization of BrdU+ cells, calculated as the fraction of the number of BrdU+ cells that co-expressed GFAP+ or Iba-1+/the total number of BrdU+ cells per section, multiplied by 100. In addition, we counted the total density population of both microglia and astrocytes, as shown in Figure 7.

### 4.8. Evaluation of Glial Transformation 

Sections containing the hippocampus were analyzed by Sholl analysis, in order to assess the number of total processes and their length, according to the methodology [84]. We quantified the number of intersections in concentric rings. We also quantified the total number and length of processes. To conduct the Sholl analysis, photographs were obtained using bright-field microscopy (Leica DMi8 microscope) at 20× magnification, connected to a DFC 7000T camera. We analyzed 120 glial cells per group/brain subregion (astrocyte and microglia) of the hippocampus.

### 4.9. Statistical Analysis

Data measurements were averaged and the significance of the differences between groups of rats were tested by one-way ANOVA and Student’s *t*-test. All statistical analysis was performed using GraphPad (GraphPad, version prism 9). Results were expressed as mean ± SEM. Post hoc test analysis (Holm–Šídák analysis to correct multiple comparisons) was employed to explore differences in single time points between control and noise (hippocampal subregions DG, CA3, CA2, and CA1) groups. Differences were considered statistically significant at a value * *p* < 0.05 (** *p* < 0.01, *** *p* < 0.001).

## Figures and Tables

**Figure 1 ijms-24-05520-f001:**
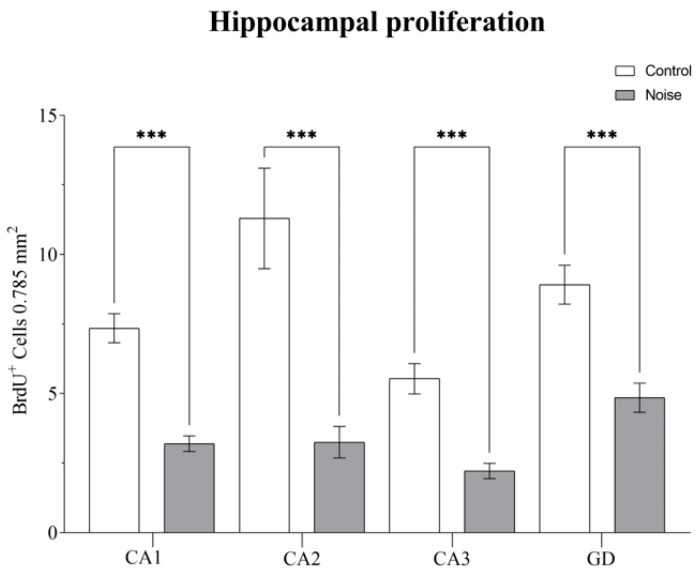
Effect of environmental noise on cell proliferation in the hippocampus. Data represent the mean ± S.E.M. One-way ANOVA was employed and considered statistically significant at a value (*** *p* < 0.001).

**Figure 2 ijms-24-05520-f002:**
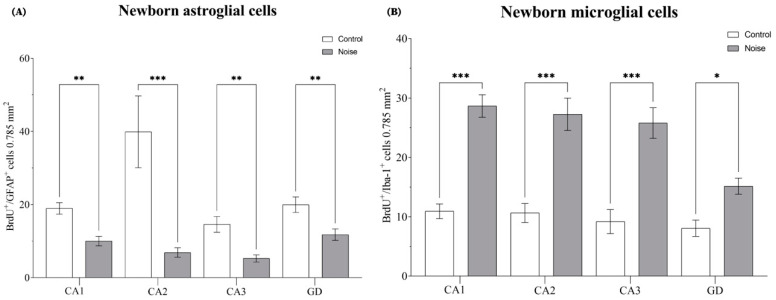
New astrocyte and microglial cells in the hippocampus. (**A**) Effect of environmental noise on the new-born astrocytes. (**B**) Represents the impact of environmental noise on the new-born microglial cells in the hippocampus. Data represent the mean ± S.E.M. One-way ANOVA was employed and considered statistically significant at a value * *p* < 0.05 (** *p* < 0.01, *** *p* < 0.001).

**Figure 3 ijms-24-05520-f003:**
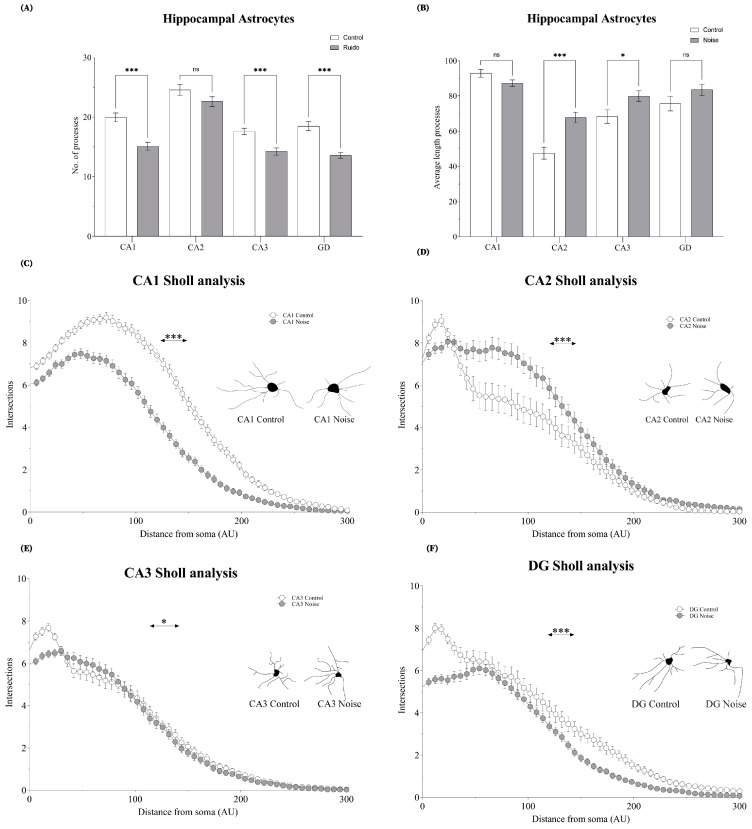
The morphometry of astrocytes in all hippocampal regions. (**A**) Displays the average number of cell processes, and (**B**) shows the average length of processes. Sholl analysis and representative astrocyte vectors are depicted in (**C**) CA1, (**D**) CA2, (**E**) CA3, and (**F**) DG. Data represent the mean ± S.E.M. *t*-student test was employed, and considered statistically significant at a value * *p* < 0.05 (*** *p* < 0.001).

**Figure 4 ijms-24-05520-f004:**
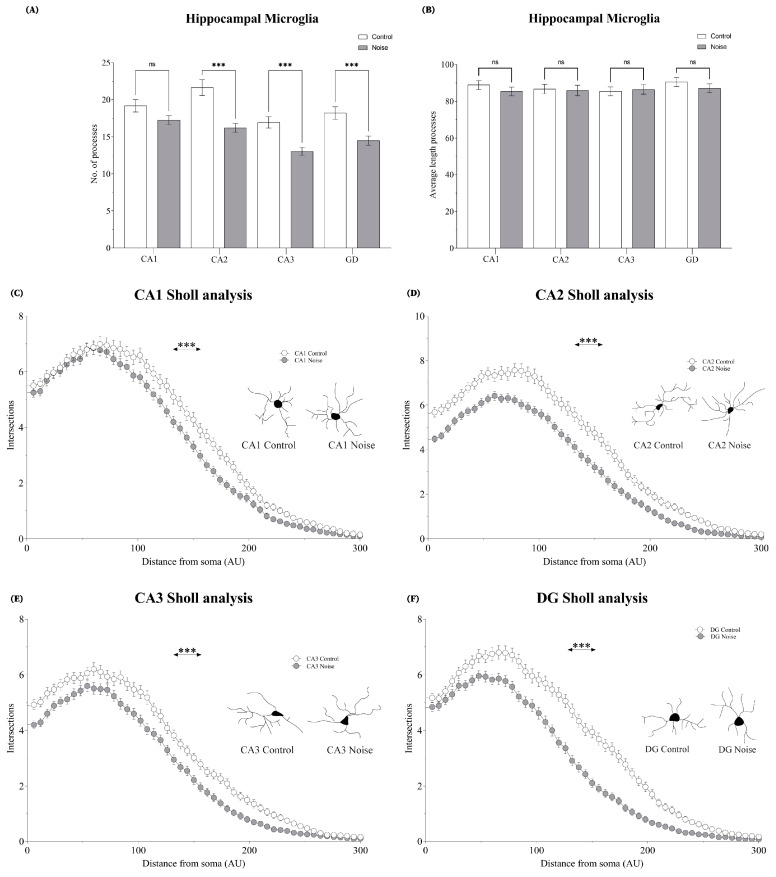
The morphometry of microglia in all hippocampal regions. (**A**) Display the average number of cell processes, and (**B**) Shows the average length of processes. Sholl analysis and representative microglia vectors are depicted in (**C**) CA1, (**D**) CA2, (**E**) CA3 and (**F**) DG. Data represent the mean ± S.E.M. Student’s *t*-test was employed and considered statistically significant at a value (*** *p* < 0.001).

**Figure 5 ijms-24-05520-f005:**
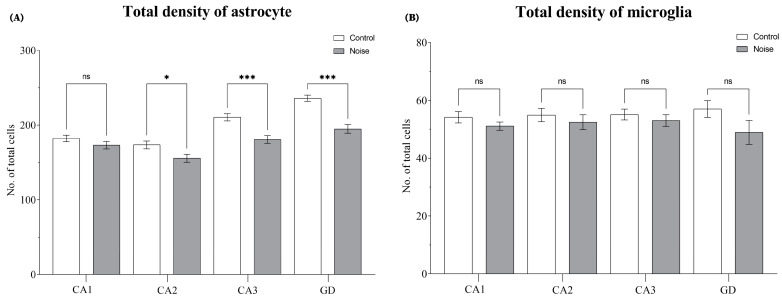
Glial population in the hippocampus. (**A**) Represents the average density of astrocytes in the hippocampus. (**B**) Represents the average density of microglia in the hippocampus. Data represent the mean ± S.E.M. One-way ANOVA was employed, and considered statistically significant at a value * *p* < 0.05, (*** *p* < 0.001).

**Figure 6 ijms-24-05520-f006:**
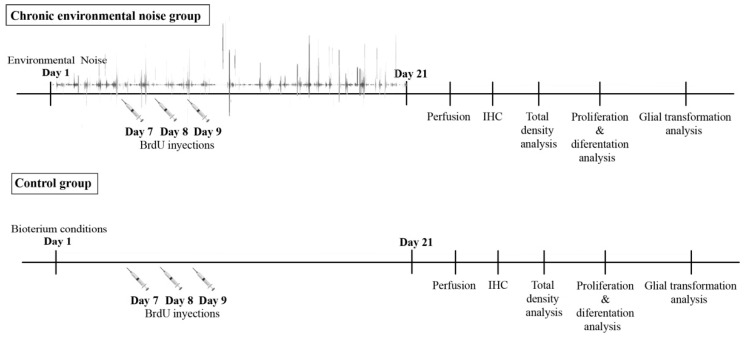
This shows the main procedures of the experiments carried out in parallel.

**Figure 7 ijms-24-05520-f007:**
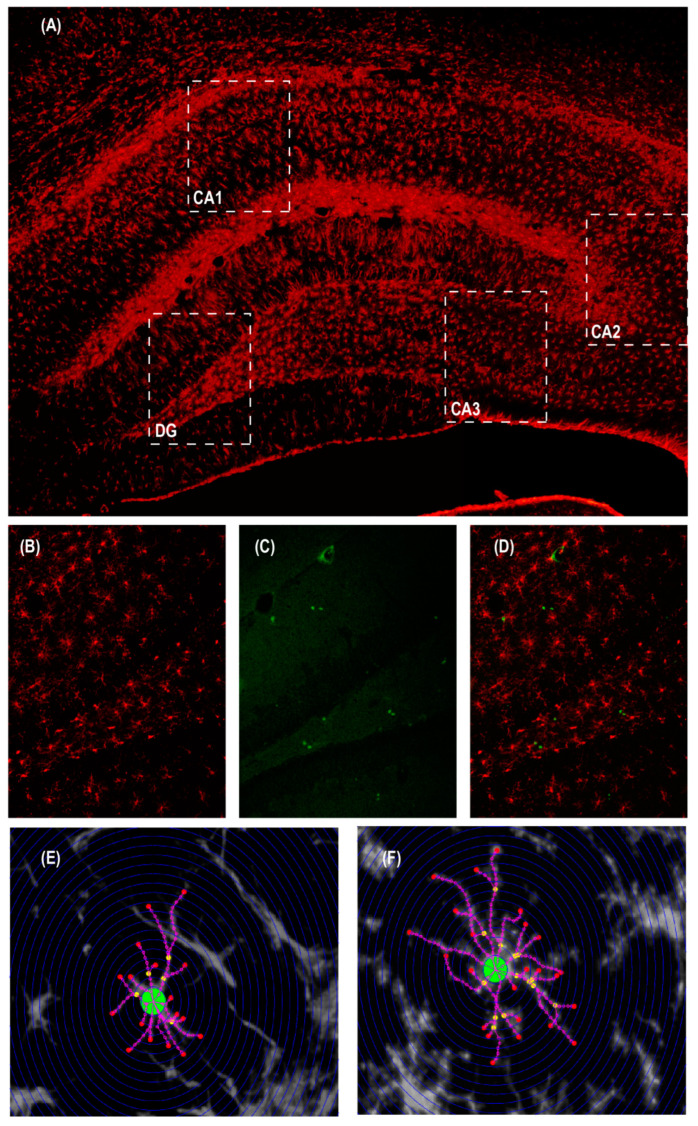
Illustrative microphotographs of the tissue staining and applied techniques. (**A**) Panoramic microphotography of hippocampal brain region, which illustrates its subregions and an approximation of the 20× microscopy field (discontinuous rectangles). In red, the GFAP stain, as an astrocyte marker, is observed. Illustrative picture made by the merging of two 5× microscope field microphotographs of the same tissue slice. (**B**) The Iba-1^+^ staining, used to identify the microglial phenotype, in red. (**C**) Microphotography (20×) illustrating, in green, the proliferation marker BrdU^+^ cells. (**D**) Illustrative merge microphotography of (**B**,**C**) images, as an example of double immunostained processing, for colocalization count. Illustrative examples of morphometric Sholl analysis on (**E**) astrocyte and (**F**) microglial cells, by zooming into a 20× microscopy field.

## Data Availability

Not applicable.

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
