# Peer review of "Acoustic Stress Induces Opposite Proliferative/Transformative Effects in Hippocampal Glia"

_ijms, 2023, doi:10.3390/ijms24065520_

Round 1

Reviewer 1 Report

Summary

This is an interesting study that investigated the effects of an environmental noise model on morphological changes of hippocampal glia, in male adult rats. Decreased proliferation of astrocytes and opposing increased proliferation of microglia was found, while both cell types showed fewer processes throughout most of the hippocampal regions.     

Major concerns

My main concern with this study regards the noise exposure protocol. Animals were intermittently exposed to 18–39 s long loud noise (70 to 105 dB) for 21 days, with only short silence intervals (20 to 165 s). This protocol would lead to chronic sleep impairment. Since sleep disturbances have already been shown to lead to morphological changes in hippocampal astrocytes (see, for example, Wadhwa et al. 2017, DOI: 10.1016/j.jneuroim.2017.09.003) and microglia (see, for example, Kincheski et al. 2017, DOI: 10.1016/j.bbi.2017.04.007), the disruption of sleep is a probable mediator in the noise exposure – altered glial morphology found here.

While sleep impairments is certainly a component of “modern stress” and “noisy urban environments”, I’m concerned that it played a major role in the environmental noise model used here. I would like to see this point addressed in the discussion.

Minor concerns

1) Unlike stated on lines 322 and 323 (Methods section), transfer to the testing room 48 h before the start of the stimuli is not apparent in Figure 13.

2) Were control animals also housed in metal grid cages like the intervention group?

3) Please correct “astrocites” to “astrocytes” in Figures 4 to 7.

Author Response

Point 1:

Major concerns

My main concern with this study regards the noise exposure protocol. Animals were intermittently exposed to 18–39 s long loud noise (70 to 105 dB) for 21 days, with only short silence intervals (20 to 165 s). This protocol would lead to chronic sleep impairment. Since sleep disturbances have already been shown to lead to morphological changes in hippocampal astrocytes (see, for example, Wadhwa et al. 2017, DOI: 10.1016/j.jneuroim.2017.09.003) and microglia (see, for example, Kincheski et al. 2017, DOI: 10.1016/j.bbi.2017.04.007), the disruption of sleep is a probable mediator in the noise exposure – altered glial morphology found here. 

While sleep impairments is certainly a component of “modern stress” and “noisy urban environments”, I’m concerned that it played a major role in the environmental noise model used here. I would like to see this point addressed in the discussion.

Response 1: We appreciate the reviewer's thoughtful comments regarding the potential impact of sleep disturbances on our findings. We acknowledge that noise exposure can lead to chronic sleep impairment, which may have played a role in the observed changes in glial morphology. This is also one of the major concerns we have addressed and discussed internally. Sleep deprivation has been acknowledged as a source of stress and other disturbances. Moreover, handling and cage environment use to be additional stressors. We would like to acknowledge the reviewer for pointing this out. It has been a source of discussion/investigation in our lab for many years. In fact, studies conducted in our lab demonstrated that sleep deprivation also induces potent stress-like responses per se. It is hard to elucidate to what extent this condition contributes to the general outcome. We believe that sleep disturbances are inherent to the noise experience. To make a realistic model, we prefer to resemble urban conditions where aircraft, road, and railway noise never stops. That makes it highly possible that noise interferes with sleep, food intake, appetite, movement, exploratory behavior, grooming, rearing, and so on. We recognize however that these variables need to be considered but, they are far beyond the scope of our current experiment.  As the reviewer can see, this is an amazing discussion that needs a special chapter. While we did not directly measure sleep in this study, we recognize that it is an important factor to consider in future research. With regards to the noise exposure protocol used here, we chose to use a range of noise levels and durations that are representative of those found in noisy urban environments. While we recognize that this may have resulted in sleep disturbances in the animals, we believe that it is a relevant and ecologically valid model for studying the effects of environmental noise on the brain. To attend the reviewer observation, we have introduced a paragraph (lines 368-377) to emphasize the stressing properties of sleep deprivation and other conditions, and to introduce the possibility of synergizing effects of these conditions.

Point 2: Unlike stated on lines 322 and 323 (Methods section), transfer to the testing room 48 h before the start of the stimuli is not apparent in Figure 13.

Response 2: We thank reviewer 2 for this observation. We would like to apologyze for this mistake. The new version of the manuscript eliminates the Figure 13 legend.

Point 3: Were control animals also housed in metal grid cages like the intervention group?

Response 2: As stated for sleep deprivation, we recognize that housing conditions could also influence the animals response to any stimuli. It is hard to make a choice on this regard. Since moving the animals to grid cages could induce aditional stress and/or stimulate exploratory behavior to the control rats, we decided to keep them in the standar cages. This is also recognized as a limitation of our study in lines 373-378.

Point 4: Please correct “astrocites” to “astrocytes” in Figures 4 to 7

Response 4: Done. Figures 4-7 were reorganized to correct these mistakes and to make it more concise.

Reviewer 2 Report

In the present manuscript Cruz-Mendoza and colleagues exposed the animals to a chronic and non-avoidant acoustic stress and investigated its impact on hippocampal cell proliferation and glial cytoarchitecture. The results showed that while after acoustic stress, there was an impaired neuronal proliferation measured by BrDU+ cells in the CA1, CA2, CA3 and DG of the hippocampus, the proliferation ratios of astrocytes and microglia showed opposite trends, both cell types displaying also morphological abnormalities compared to control animals.

Collectively, the manuscript is well written and presented. Experimental design and analysis seem properly done.

I have only some minor questions/comments:

·       Was the noise regime on during 24h/day?

·       Did you take blood samples at the end of the 21 days to measure corticosterone or any other marker of stress to confirm that the animals were stressed?

·       Why in Figure 14E and F the radius of the concentric rings has not the same distance?

·       If the BrDU is administered on day 7, 8 and 9 and the noise exposure is continued until day 21, are not the authors measuring here survival instead of cell proliferation?

·       There is small typo in figures 4,5,6 and 7 in the word astrocyte. Same in line 169

·       It would help to add a legend in the bar figures 1 and 2 just indicating that the white bars are the control, and the grey are the noise.

·       The density of astrocytes and microglia figure should be 13 instead of 3 therefore the following figures need different numbers. Also, there is no figure 3, so all the figures need new numeration.

·       In the discussion section (line 225-226), where the authors write “It is important to note that some of these cells, specifically those located in the subgranular layer of the dentate gyrus, are b-precursor cells, and that the specific decrease of these cells affects the replenishment of precursors in one of the main neurogenic niches of the adult brain” Which neurogenic niche are you referring?

·       In lines 245-248 “Although 245 astrocyte transformation is closely related to post-lesion damage, its change could also 246 reflect a mechanism that allows the hippocampus to prioritize certain functions and not 247 necessarily a failure by itself”. Do you have any reference for that?

Author Response

Point 1: Was the noise regime on during 24h/day?

Response 1: Yes. For this study, we employed a 24h/day regime

Point 2: Did you take blood samples at the end of the 21 days to measure corticosterone or any other marker of stress to confirm that the animals were stressed?

Response 2: No. We do not take any blood samples in this specific study. We have already conducted a number of experiments confirming that environmental noise as used here, induce strong elevations in serum corticosterone levels. Some references to sustain this can be viewed at:

Ruvalcaba-Delgadillo Y, Luquín S, Ramos-Zúñiga R, Feria-Velasco A, González-Castañeda RE, Pérez-Vega MI, Jáuregui-Huerta F, García-Estrada J. Early-life exposure to noise reduces mPFC astrocyte numbers and T-maze alternation/discrimination task performance in adult male rats. Noise Health. 2015 Jul-Aug;17(77):216-26. doi: 10.4103/1463-1741.160703. PMID: 26168952; PMCID: PMC4900483.

Gonzalez-Perez O, Chavez-Casillas O, Jauregui-Huerta F, Lopez-Virgen V, Guzman-Muniz J, Moy-Lopez N, Gonzalez-Castaneda RE, Luquin S. Stress by noise produces differential effects on the proliferation rate of radial astrocytes and survival of neuroblasts in the adult subgranular zone. Neurosci Res. 2011 Jul;70(3):243-50. doi: 10.1016/j.neures.2011.03.013. Epub 2011 Apr 15. PMID: 21514330.

Jáuregui-Huerta F, Garcia-Estrada J, Ruvalcaba-Delgadillo Y, Trujillo X, Huerta M, Feria-Velasco A, Gonzalez-Perez O, Luquin S. Chronic exposure of juvenile rats to environmental noise impairs hippocampal cell proliferation in adulthood. Noise Health. 2011 Jul-Aug;13(53):286-91. doi: 10.4103/1463-1741.82961. PMID: 21768732.

Point 3: Why in Figure 14E and F the radius of the concentric rings has not the same distance?

Response 3: We thank the reviewer for pointing this. It was an involuntary mistake occurred during the figure construction process. It was corrected in the new version of the manuscript.

Point 4: If the BrDU is administered on day 7, 8 and 9 and the noise exposure is continued until day 21, are not the authors measuring here survival instead of cell proliferation?

Response 4: We thank the reviewer for calling our attention on this. We also believe that using “proliferation” could be confuse. However, the same apply for “survival” since it can refer all the surviving cells in the hippocampus (all the living cells which generally are demonstrated by tunel or other apoptotic tecnniques). In any case, we are including here the cells that proliferate at days 7,8 and 9 (incorporate the injected BrdU) and lived to be counted at day 21 (new-born cell survival). It is important to keep in mind that waiting these two weeks is mandatory to observe differentiated cells (i.e. astrocytes and microglia).

Point 5: There is small typo in figures 4,5,6 and 7 in the word astrocyte. Same in line 169

Response 5: Thanks to reviewer 2 for pointing this. It was corrected in the new version of the manuscript.

Point 6:  It would help to add a legend in the bar figures 1 and 2 just indicating that the white bars are the control, and the grey are the noise.

Response 6: Thanks to reviewer 2 for pointing this. It was also corrected in the new version of the manuscript.

Point 7: The density of astrocytes and microglia figure should be 13 instead of 3 therefore the following figures need different numbers. Also, there is no figure 3, so all the figures need new numeration.

Response 7: Thanks to reviewer 2 for pointing this. Figures were reorganized in order to correct these mistakes and make the results more clear and concise.

Point 8: In the discussion section (line 225-226), where the authors write “It is important to note that some of these cells, specifically those located in the subgranular layer of the dentate gyrus, are b-precursor cells, and that the specific decrease of these cells affects the replenishment of precursors in one of the main neurogenic niches of the adult brain” Which neurogenic niche are you referring?

Response 7: We agree with reviewer 2 on this point. It can be confuse as it was stated. Lines 305-306 of the new version clarify this.

Point 9: In lines 245-248 “Although 245 astrocyte transformation is closely related to post-lesion damage, its change could also reflect a mechanism that allows the hippocampus to prioritize certain functions and not necessarily a failure by itself”. Do you have any reference for that?

Response 7: We thank the reviewer 2 for this observation. This can be sustained by references 55, 56 that analyze the physiological/pathological sense of glial transformation. References have been already included in line 328.